# Are the Parents’ and Their Children’s Physical Activity and Mode of Commuting Associated? Analysis by Gender and Age Group

**DOI:** 10.3390/ijerph17186864

**Published:** 2020-09-20

**Authors:** Fernando Rodríguez-Rodríguez, Francisco Javier Huertas-Delgado, Yaira Barranco-Ruiz, María Jesús Aranda-Balboa, Palma Chillón

**Affiliations:** 1IRyS Group, School of Physical Education, Pontificia Universidad Católica de Valparaíso, 2374631 Valparaíso, Chile; 2Teacher Training Centre La Inmaculada, University of Granada, 18003 Granada, Spain; fjhuertas@ugr.es; 3PROFITH “PROmoting FITness and Health through Physical Activity” Research Group, Department of Physical Education and Sport, Faculty of Sport Sciences, University of Granada, 18001 Granada, Spain; ybarranco@ugr.es (Y.B.-R.); mjab@ugr.es (M.J.A.-B.); pchillon@ugr.es (P.C.)

**Keywords:** active transport, active commuting to school, school, schoolchildren, family

## Abstract

Background: Some studies have reported a positive parent–child association between physical activity (PA), but few have examined the difference in these associations concerning both genders. The objective of this study was to establish the association between moderate to vigorous physical activity (MVPA) and mode of commuting (MC) of the parents with their children by gender and age group. Methods: This cross-sectional study included 686 mothers and fathers (43.4 ± 6.5 years old) and their children (children 9.7 ± 1.7 y. and adolescents 14.0 ± 1.7 y.). Each participant completed a questionnaire on PA and MC. Chi-square test, odds ratio for categorical variables, and lineal regressions for continuous variables were used to examine the parent–child associations. Results: An inverse association was found between fathers–children in the weekend MVPA in children and between mothers–adolescents in out-of-school and weekend MVPA. An inverse association was found in MVPA between mothers-girls, and the different parents’ MC to work was positively associated with the MC to school in children and adolescents except for the association AC parents–adolescents. The AC was mainly associated between mothers and girls and boys. Conclusions: A weak association in parent–child MVPA but a strong association in MC between parent–child was found.

## 1. Introduction

Regular physical activity (PA) has been associated with numerous health benefits at all stages of life, especially in youth [1,2]. To obtain these benefits, it is recommended to perform at least 60 min/day of moderate to vigorous PA (MVPA) in young people and at least 150 min/week of MVPA in adults [3,4]. Unfortunately, a small proportion of children/adolescents [5,6] and adults [7] meet these recommendations. On the other hand, the studies have shown overall that boys and children are more active than girls and adolescents, respectively [8].

Children’s PA habits are shaped by their parents, particularly in younger children compared with adolescents [9]. It has been estimated that children with active parents are three to four times more active than children with inactive parents [10]. In addition, a greater PA relationship with children compared to adolescents has been demonstrated [11].

Moreover, the effect of the parent gender on children’s PA has been previously examined. For instance, a positive association has been found between mothers’ sport participation and children’s leisure time PA, but it remained significant only in girls [12]. Another Brazilian study observed a greater positive association between the mother’s PA with adolescent boys and girls, but there was no association for fathers [13]. Furthermore, the PA of both parents presented a greater association with girls than with boys. In contrast, a study demonstrated that only the mother’s MVPA was associated with their children’s MVPA [14]. Another study showed a significant positive association between fathers and boys only [15]. Despite that fact, studies have focused on the mother–child relationship, and relatively little attention has been paid to the role of fathers in their children’s PA [16,17]. Mothers play a greater role in planning and organizing children’s PA, while fathers are more likely to model children’s PA [18]. Until now, the evidence on PA gender-specific parental influence (e.g., mother–daughter, mother–son, father–daughter, father–son) is still inconsistent. 

It has been established that active commuting to school (ACS) or active commuting (AC) to work (walking or cycling) is an opportunity to increase PA levels [19,20]. Moreover, ACS is related to nine sustainable development goals, emphasizing the third (health and well-being) and the eleventh (sustainable cities and communities) [21]. However, around 50% of youths and 75% of adults passively commute to school [22] and to work [23], respectively. Multiple factors influence whether schoolchildren actively travel to/from school [24]. The educational level and the body weight [21], the socioeconomic level [25], and the unemployment of the parents [26] are factors that affect children’s ACS and have been studied. However, few studies have linked the parent’s AC with their children’s ACS, and even fewer take into account gender. A Spanish study observed a positive association between parents’ AC and children’s ACS [26]. Indeed, the number of steps/day of the parents has been positively associated with the number of steps/day of the children [27], mainly associated between mothers and daughters [28]. Specifically, it has been reported that, for every 1000 steps/day that parents increase, their children increase by 260 steps/day [29]. Therefore, further quality studies analyzing the influence that the fathers’ or the mothers’ AC has on boys’ or girls’ ACS as well as taking into account the age (e.g., children or adolescents) are needed. In the current study, it is expected to verify the association between parent–child PA and to find new associations between the AC that support the importance of the family in the development of active behaviors. This would allow targeting interventions to more specific sectors and groups in order to achieve successful results. 

According to the previous information, the main objective of this study was to establish the association between moderate to vigorous physical activity (MVPA) and mode of commuting (MC) of the parents with their children by gender and age group.

## 2. Materials and Methods 

### 2.1. Study Design and Participants

It is a cross-sectional study with children and parent’s participation carried out in Granada (Spain) and Valparaíso (Chile). Data were obtained as part of the “Cycling and Walk to School” (PACO, for its Spanish acronym) study, focused on promoting PA and, particularly, active commuting to and from school. A total of 2526 children and adolescents and 1959 of their parents participated. Twenty schools were invited to participate in the study as a non-randomized sample. From the total children sample, 1807 participants could not be paired with parents’ data, and 34 did not report their gender and were excluded (72.8% of the total sample). A total of 572 parents could not be paired with children’s data, and 703 without gender data were excluded (65% of the total sample). Finally, a total of 686 paired parents (52.8% mothers) and their respective children (33.7% girls) were considered. They belonged to fifteen schools of Granada Spain (*n* = 494) and five schools of Valparaíso in Chile (*n* = 192). The age (mean ± standard deviation) of each group was: parents 43.4 ± 6.5 years old), children 9.7 ± 1.7 years old, and adolescents 14.0 ± 1.7 years old.

### 2.2. Sociodemographic Factors

Children and adolescents completed a questionnaire on sociodemographic characteristics, which includes school name, age, birth date, school grade and class, gender, birth country, mother’s and father’s birth country, and full postal address. Parents completed a questionnaire on sociodemographic variables such as age, gender, monthly income (none; <499€; 500–999€; 1000–1499€; 1500–1999€; 2000–2499€; 2500–2999€; 3000–4999€; >5000€) and highest educational level (no study, primary school, secondary school, bachelor, professional, university degree). The monthly income was dichotomized in ≤999€ or ≥1000€, and the highest educational level was dichotomized in secondary/bachelor’s (no study, primary school, secondary school, bachelor) or professional/university (professional or university degree).

### 2.3. Physical Activity

The questionnaire to determine the weekly school and extracurricular PA in children and adolescents was the Youth Activity Profile (YAP), that was self-reported by children and adolescents. The YAP provides a simple and low-cost method that has already been calibrated and validated to accurately estimate children’s MVPA and sedentary behaviors at the group level [30]. The self-reported YAP questionnaire comprises three items about participation in different types of activities and sports in the last seven days. The items were “PA in school” (physical education, recess, and ACS), “out-of-school PA” (before and after school and PA at the weekend), and “sedentary time” (not considered in current study). Each item was scored on a scale from 1 (low PA) to 5 (high PA), and the average score denoted the YAP score (1–5). The questions were categorized into three areas: PA at school, out-of-school PA, and weekend PA. Then, PA in children and adolescents was established by transforming the YAP score to minutes/day in MVPA using the Fairclough equations [31]. We obtained the min/day in MVPA at school, out-of-school MVPA, and weekend MVPA separately for children and adolescents and for boys and girls. The cut-off point to be “physically active” was >60 min/day MVPA [32], classified as meeting MVPA recommendations. Those children who did not comply were considered as “physically inactive”.

The International Physical Activity Questionnaire (IPAQ, short version) was used to assess the parents’ MVPA. This questionnaire has been validated in 12 countries [33,34], showing acceptable psychometric properties to measure the MVPA levels in one week. The IPAQ allows determining the PA in min/week. Additionally, this instrument determines intensity categories as sedentary time, light PA, moderate PA, and vigorous PA. According to the MVPA international recommendations, parents were classified as physically active when they completed ≥150 min/week (meeting the recommendations) and physically inactive when they did not reach 150 min/week (not meeting the recommendations).

### 2.4. Mode of Commuting

The instruments used to assess the mode of commuting were a student (for children and adolescents) and a family (for parents) questionnaire from the PACO Study that have been developed at the University of Granada, Spain by a group of experts in this topic [35]. The questions about the MC to school derive from an exhaustive review of previous studies of the scientific literature on AC [36] and have been reliable [37] and validated in Spanish population [38] and validated in Chilean population [39]. In children and adolescents, the questions included in the questionnaire were: (1) “How do you usually get to school?” and (2) “How do you usually get home from school?”. The possible answers were: walking, cycling, car, motorcycle, school bus, public bus, metro/train, and other (the mode description was required). The final variable to analyze was usual active mode of commuting to school.

The questions about parent’s MC to work have gone through an exhaustive reliability process [40]. In parents, the questions were: (1) “How do you usually get to work?” and (2) “How do you usually get home from work?”. The possible answers were: walking, cycling, car, motorcycle, public bus, metro/train, or other (the mode description was required). 

MC for children and parents was categorized as “active” (walking and cycling) and “passive” (car, motorcycle, bus, metro/train). Additionally, the “passive” commuting was divided into private mode of commuting (car, motorcycle) and public mode of commuting (public bus, metro, train). In addition, the final variable of usual active commuting included one or two trips to school (to, from, or both).

### 2.5. Procedures

The questionnaires (paper-and-pencil) were administered at participating schools between 2015 and 2018. The children and the adolescents’ questionnaires were implemented by the researcher staff during the school hours within the physical education lessons and approximately by 30 min. Both research team and schoolteachers presented this questionnaire for clarification purposes. The family’s questionnaire was completed once by parents, and it was delivered to children and completed at home by parents.

Parents signed an informed consent that described the objectives and the characteristics of this study and allowed their children to participate, in accordance with the Declaration of Helsinki. This study was reviewed and accepted by both the Ethical Committee of the University of Granada Spain (No.162/CEIH/2016) and the Ethical Committee of the Pontificia Universidad Católica de Valparaíso, Chile (CCF02052017).

### 2.6. Statistical Analysis

Descriptive statistics were calculated for study variables, mean (M) and standard deviation (±SD) for continuous variables and frequency (%) for categorical variables. The comparison of categorical variables according to gender in each group (parents (mothers and fathers), children (girls and boys), adolescents (girls and boys)) was examined using the Chi-Square test.

To establish the association between parent–child’s MVPA, several lineal regressions through a standardized coefficient (ß) were used for the whole sample for boys and for girls separately. The min/day of MVPA in children and the dichotomic variable of meeting the recommendations for parents were included. Children’s MVPA was established as the dependent variable. For the MC association, several binary logistic regressions were performed. Children’s MC to school was established as the dependent variable. The references for MC were (a) “passive” for active commuting; (b) “passive + private” for public commuting; (c) “passive + public” for private commuting. The associations were adjusted for parents’ educational level and age of the children. IBM SPSS^®®^ v21 (New York, NY, USA) was used for all the analyses. A *p* < 0.05 value was established as statistically significant.

## 3. Results

Parent–child sociodemographic characteristics are presented in Table 1. The children and the parents’ MVPA and MC separated by gender are presented in Table 2. The percentage of physically inactive girls was significantly higher than boys for both children (*p* < 0.001) and adolescents (*p* = 0.003). Parents of children and parents of adolescents were also mainly physically inactive (75% and 84.9%, respectively) in both genders (mother and father). 

The associations between the parents’ MVPA and their offspring’s MVPA by gender are presented in Table 3. In the children group, only the fathers’ MVPA was positively associated with girls’ MVPA (ß = 0.24; 95% CI: 0.00–1.80) and negatively associated with boys’ MVPA (ß = −0.27; 95% CI: −2.10–−0.19), both of them at the weekend. In adolescents the mothers’ MVPA was negatively associated with the MVPA in all out-of-school (ß = −0.23; 95% CI: −5.12–0.27) and in girls during the weekend (ß = −0.38; 95% CI: −2.35–−0.40). 

The associations between the parents’ MC to work and their children MC by gender are presented in Table 4. Mothers’ AC was associated with girls’ and boys’ ACS in children (OR = 4.48; 95% CI: 2.01–9.96 and OR = 5.19; 95% CI: 1.92–14.05, respectively). Fathers’ AC to work was only associated with the girls’ ACS (OR = 3.69; 95% CI: 1.83–7.47). Girls presented higher odds to use public commuting if fathers used public commuting. Public commuting was only associated between fathers and girls (OR = 12.62; 95% CI: 2.08–76.55). Mothers of children who used private commuting increased the likelihood that girls and boys also used private commuting (OR = 2.24; 95% CI: 1.32–3.80 and OR = 6.26; 95% CI: 2.84–13.81, respectively). Children had higher odds to use private commuting when fathers used private commuting (girls OR = 3.08; 95% CI: 1.83–5.16 and boys OR = 7.96; 95% CI: 3.48–18.20). In adolescents, no associations were found between parents’ AC to work and children ACS. In fathers, an association with girls was observed in public modes (OR = 6.30; 95% CI: 2.21–17.94) as well as with mothers (OR = 4.59; 95% CI: 1.76–11.94). Additionally, when mothers used private commuting, boys presented higher odds to use private commuting (OR = 17.21; 95% CI: 2.60–113.70), and when fathers used private commuting, girls showed higher odds to use private modes of commuting (OR = 12.72; 95% CI: 5.01–32.31).

## 4. Discussion

The main findings of the study were: (1) an inverse association was found between fathers–children in the weekend MVPA in children and between mothers–adolescents in out-of-school and weekend MVPA in adolescents, specifically, an inverse association was found in MVPA between mothers-girls; and (2) the different parents’ MC to work were positively associated with the MC to school in children and adolescents except for the association AC parents–adolescents and, specifically, the AC was mainly associated between mothers and girls and boys.

### 4.1. Physical Activity

In children, the fathers’ MVPA was only positively associated with the girls’ weekend MVPA. In addition, an inverse association in MVPA between mothers–boys in children, mother–child out-of-school MVPA and mothers–girls in the weekend PA in adolescents was found. 

Accordingly, a modest and positive association between parent–child PA in a systematic review was found (52% of studies), but no differences were observed in the influence of mothers or fathers on children’s PA [16]. However, previously a positive association was found for fathers–boys and mothers–girls PA [41], but the stronger correlation was observed for father–boy PA, and there was a lighter association for mothers–boys and mothers–girls PA [42]. 

Our results in children showed that fathers’ MVPA was positively associated with the weekend MVPA in girls and negatively associated with boys. The positive association between active fathers and active girls can be explained by the role modeling of parents for girls’ PA, especially at the weekend. For example, girls whose parents exercised ≥3 times weekly could be almost 50% more active than girls with sedentary parents. This especially occurred at the weekend where there was a greater interaction between parents and children [43]. In this regard, it has been reported that the fathers were more likely to offer social support to practice PA than mothers, whose main role is taking care of their children during the weekdays because fathers work long hours during the day [44]. Furthermore, the negative association fathers–boys found in the current study can be explained by the low number of parents who met the PA recommendations. This makes it difficult to find clear associations. Additionally, we assume that fathers are not role modeling to children PA but support the practice [45,46].

Our results in adolescents showed that the mother’s MVPA was negatively associated with out-of-school MVPA in both (girls and boys) and with girls’ MVPA during the weekend. The other parents´ associations with boys and girls were not significant. These negative associations can be explained because the adolescents’ PA is more independent of the parents’ PA. This can be especially accentuated because mothers comply with the MVPA recommendations less than fathers. Additionally, children, as they grow up, can have active behavior independent of their parents and more related to their peers [47]. In this regard, it has been reported in adolescents that paternal and maternal support decreased with age, whereas independent play increased [48]. In addition, a qualitative study showed that girl adolescents received more emotional negative support for PA from parents and less tangible support and direct participation from parents in PA with them [49]. It has been found that it may be that sedentary girls are likely to have sedentary parents, while there may be a weak association between parents and regularly active girls [50].

In general terms, in the current study, no positive association parent–child PA was found. In this regard, a recent systematic review showed a weak correlation between the level of parent and child PA and suggests that we should develop a deeper understanding of associations and mechanisms [51]. Future studies and interventions should investigate the association between parents and their children PA, especially out-of-school and at the weekend, which seems to be the time where there is a greater association. 

### 4.2. Mode of Commuting

In the current study, several positive associations between parents and children in the different MC (active, public, and private) were found. A greater association was found between mothers’ AC to work with girls’ and boys’ ACS than fathers. The fathers’ AC to work was only associated with the ACS in girls. This finding could be related to a greater degree of care or responsibility assumed by mothers with their children [52], where mothers were mainly responsible for taking them to school. Another study found that greater mothers’ work flexibility was related to a greater number of children’s active trips [53]. Furthermore, it was possible to associate the MC of the mothers, but not the fathers, with the ACS of their children [54].

In our study, the association between parents’ AC to work and children ACS was stronger in children than adolescents. This could be explained by the higher degree of autonomy and independence of adolescents who no longer depend heavily on parents for commuting [11]. In addition, it has been studied that the parents perceived lower barriers in adolescents than children [55,56,57]. These factors would cause adolescents to choose a different MC than their parents because they independently commute to school empowered by social interaction among peers [58].

The general results of the current study showed a positive association between public commuting of parents and girls. However, no associations between mothers–boys or fathers–boys were observed. These results could be explained by a possible greater independence of boys compared to girls, which has been previously reported [59]. Girls would prefer to be accompanied by their parents, or parents impose more barriers for independent commuting to girls [60]. Using public transport is important, since, along with walking, they are considered the two best options for independent mobility in children and adolescents [61]. Previous studies have indicated that parents feel more secure and less afraid to allow boys to walk further from home than girls [62,63]. This would affect girls’ ACS and increase the use of passive MC.

In the current study, a positive association was observed between the parents’ private MC and their children MC. This association was even higher than the observed with AC, especially among parents and adolescents. A greater positive association between mothers and boys’ adolescents in private commuting was found. Likewise, a strong association between fathers with adolescents’ girls in this mode was observed. In this regard, a Spanish study in schoolchildren of 9–12 years [64] observed an inverse association between the fact that it is more convenient to drive and the accompaniment of their children on foot to school. Parents’ convenience of using the car to carry their children has been previously reported as one of the main barriers to ACS [65,66]. In addition, in the case of adolescents, it has been reported that they prefer to be driven to school [67] than another MC. The main problem is that these preferences of young people to use passive MC could be transferred to adult life [64]. The findings from the current study about passive commuting are relevant, since not only are children more likely to use ACS if parents use AC, but also children are still more likely to use passive MC when parents do too.

### 4.3. Active Families

The associations in MVPA and MC could also be bi-directional, that is, they can benefit parents and children [68]. Accordingly, we propose a classification of families according to whether they have reached the MVPA recommendations and according to their use AC (Figure 1). It is important to note that compliance with the MVPA recommendations has been considered as a behavior that is above the AC to be considered active [69]. Therefore, active commuting is a secondary factor. However, the contribution that the AC has as a means to achieve the recommendations cannot be ignored either. According to the classification, our proposal defines “Active Family” as those parents and children that meet the MVPA recommendations and use AC. “Moderately Active Family” is applied to those who use AC, but parents or children do not meet MVPA recommendations. In this case, the categorization is affected when any member does not meet the MVPA recommendations. The category “Inactive Family” is applied to the families where neither parents nor children meet the MVPA recommendations. It makes the entire group inactive, even though some member actively commutes. Finally, “Inactive and Passive Family” includes the families where parents and children do not meet MVPA recommendations and use passive commuting. 

For a better interpretation, it should be considered that other members may positively or negatively affect PA or AC (grandparents, siblings, uncles, etc.). Therefore, by pointing to “parent”, it may refer to any parental figure influencing children.

### 4.4. Limitations

The main limitation of the study was the cross-sectional design and, therefore, no cause and effect relationship can be established in the associations found. A longitudinal study would be required to determine the direction of the relationship. There was a relevant loss data of sample regarding the initial data collection, because the questionnaires were incomplete. Additionally, a non-randomized sample was included; therefore, it is not possible to generalize results to the population. In addition, a self-reported questionnaire that has a lower objectivity to determine the PA than devices such as accelerometers was used.

Finally, in relation to the mode of commuting, we did not have the distance to and from school (continuous value), and it was not used to establish the impact on AC.

### 4.5. Study Application and Projections

The findings allow targeting interventions and programs at school to promote PA and ACS towards girls and their mothers, which is where there is a greater association. Literature shows that mothers and girls are the ones who do less PA. Thus, promoting ACS in girls would help increase the time dedicated to daily PA. In addition, it is necessary that parents be involved in school activities for the promotion and the practice of PA and to promote healthy behaviors in the educational community. Future studies should include an objective evaluation of PA and ACS that allows one to be precise in the associations found. However, a questionnaire to determine sociodemographic and implicated variables should continue to be applied. Some variables that can be included are the length of the parents’ working hours and determining if the siblings play an important role in the association over the parents.

## 5. Conclusions

The results in the current study demonstrate a weak association between the parent–child MVPA. This lack of association could be related to gender stereotypes, family structure (single parent, extended or reconstituted family, etc.), or social problems, among other factors. More studies should be conducted to further explore the relationships and the associations between parent–child MVPA in different countries and realities. Otherwise, a strong association was found between the MC to work of parents and the MC to school of their children. It was not possible to establish a clear association of the fathers’ AC over the mother or vice versa about ACS of their boys and girls. Increasing the parents’ AC to work could mean an improvement not only in their own levels of MVPA but also in the MVPA levels and the ACS of their children. Otherwise, attention should be given to the use of passive modes of parents, which are largely associated with the passive modes in children. The results suggest the importance of involving families in the design, the implementation, and the evaluation of interventions in PA and ACS in children and adolescents. 

## Figures and Tables

**Figure 1 ijerph-17-06864-f001:**
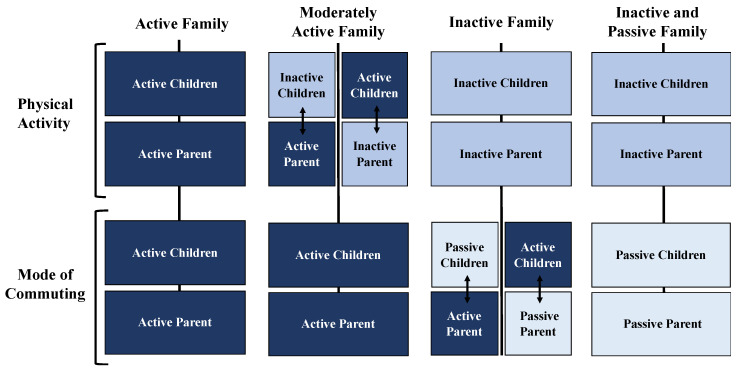
Proposal for classification of families according to compliance with MVPA recommendations and the use of active MC.

**Table 1 ijerph-17-06864-t001:** Sociodemographic characteristics of the parents of children and parents of adolescents.

	Overall		Parents of Children		Parents of Adolescents		*p*-Value
	N	(%)	N	(%)	N	(%)	
**Children gender (*n* = 686)**							
Girls	232	(33.8)	153	(34.9)	79	(32.0)	0.249
Boys	454	(66.2)	286	(65.1)	168	(68.0)	
**Children age (M ± SD)**	11.3 ± 2.7		9.7 ± 1.7		14.0 ± 1.7		** <0.001
**Parent gender (*n* = 686)**							
Mothers	362	(52.8)	245	(55.8)	118	(47.7)	* 0.026
Fathers	323	(47.2)	194	(44.2)	129	(52.3)	
**Parent age (Mean ± SD)**	43.4 ± 6.5		41.9 ± 5.7		46.7 ± 7.1		** <0.001
**Mother’s highest educational level (*n* = 614)**							
Secondary school/bachelor’s	330	(53.7)	194	(49.7)	136	(60.7)	* 0.009
Professional/university	284	(46.3)	196	(50.3)	88	(39.3)	
**Father’s highest educational level (*n* = 675)**							
Secondary school/bachelor’s	314	(46.5)	190	(43.9)	124	(51.2)	0.066
Professional/university	361	(53.5)	243	(56.1)	118	(48.8)	
**Mother’s monthly income (*n* = 336)**							
<1000€	127	(37.8	75	(33.0)	52	(47.7)	* 0.009
>1000€	209	(62.2)	152	(67.0)	57	(52.3)	
**Father’s monthly income (*n* = 463)**							
<1000€	278	(60.0)	183	(57.0)	95	(66.9)	* 0.045
>1000€	185	(40.0)	138	(43.0)	47	(33.1)	

Statistical differences in Chi-square * *p* < 0.05; ** *p* < 0.001.

**Table 2 ijerph-17-06864-t002:** Descriptive data of parents’ and their children physical activity and mode of commuting by gender.

	Children						
		Children			Adolescents		
	Overall	Girls	Boys	*p*-Value	Girls	Boys	*p*-Value
	N (%)	N (%)	N (%)		N (%)	N (%)	
**MVPA ^a^**							
Physically active	185 (33.3)	62 (25.0)	111 (88.8)	<0.001 **	4 (3.1)	8 (15.1)	0.003 *
Physically inactive	370 (66.6)	186 (75.0)	14 (11.2)		125 (96.9)	45 (84.9)	
**MC to school**							
Active	220 (32.1)	89 (31.1)	50 (32.7)	0.738	52 (31.0)	29 (37.2)	0.333
Passive	465 (67.9)	197 (68.9)	103 (67.3)		116 (69.0)	49 (62.8)	
	**Parents**						
		**Parents of Children**			**Parents of Adolescents**		
	**Overall**	**Mother**	**Father**	***p*-Value**	**Mother**	**Father**	***p*-Value**
	**N (%)**	**N (%)**	**N (%)**		**N (%)**	**N (%)**	
**MVPA ^a^**							
Physically active	49 (7.2)	7 (2.9)	28 (14.4)	<0.001 **	3 (2.6)	11 (8.5)	0.044 *
Physically inactive	635 (92.8)	217 (97.1)	166 (85.6)		114 (97.4)	118 (91.5)	
**MC to work**							
Active	216 (31.6)	81 (33.2)	51 (26.4)	0.126	44 (37.6)	40 (31.0)	0.276
Passive	467 (68.4)	163 (66.8)	142 (73.6)		73 (62.4)	89 (69.0)	

MVPA; moderate to vigorous physical activity; MC: mode of commuting. Statistical differences in Chi-square * *p* < 0.05; ** *p* < 0.001. ^a^ Presents missing data in sample.

**Table 3 ijerph-17-06864-t003:** Associations between parents’ MVPA and their children physical activity by gender.

	MVPA Mother		MVPA Father			MVPA Mother		MVPA Father	
	*ß*	95% CI	*ß*	95% CI		*ß*	95% CI	*ß*	95% CI
Children					Adolescents				
**MVPA total**					**MVPA total**				
All	−0.12	(−3.95–0.37)	−0.07	(−3.60–1.46)	All	−0.14	(−10.30–2.32)	0.01	(−7.20–7.55)
Girls	−0.03	(−1.97–1.43)	0.01	(−2.08–2.17)	Girls	−0.11	(−7.88–3.47)	0.19	(−3.18–11.02)
Boys	0.03	(−1.81–2.48)	−0.10	(−3.59–1.48)	Boys	−0.02	(−10.53–9.51)	0.40	(−0.12–17.83)
**MVPA in school**					**MVPA in school**				
All	−0.10	(−3.09–0.57)	−0.02	(−2.40–1.92)	All	−0.04	(−5.74–3.92)	−0.06	(−6.50–4.03)
Girls	−0.06	(−1.41–0.75)	0.04	(−1.23–1.72)	Girls	−0.01	(−3.81–3.46)	0.84	(−2.54–6.17)
Boys	0.02	(−1.30–1.62)	−0.06	(−2.31–1.31)	Boys	0.17	(−4.17–10.29)	0.39	(−0.28–10.92)
**Out-of-school MVPA**					**Out–of–school MVPA**				
All	0.01	(−0.65–0.76)	−0.15	(−0.61–0.00)	All	−0.23 **	(−5.12–0.27)	0.17	(−0.90–4.95)
Girls	0.02	(−0.90–1.06)	−0.08	(−1.61–0.82)	Girls	−1.17	(−4.79–1.11)	0.18	(−1.54–5.67)
Boys	0.06	(−0.72–1.33)	−0.09	(−1.65–0.64)	Boys	−0.27	(−8.57–1.81)	0.24	(−1.54–8.96)
**MVPA at the weekend**					**MVPA at the weekend**				
All	−0.08	(−1.27–0.31)	−0.06	(−1.36–0.58)	All	−0.20	(−2.53–0.05)	−0.15	(−2.42–0.63)
Girls	−0.10	(−1.01–0.33)	0.24 **	(0.00–1.80)	Girls	−0.38 **	(−2.35–0.40)	0.01	(−1.32–1.40)
Boys	0.04	(−0.66–1.00)	−0.27 **	(−2.10–0.19)	Boys	0.01	(−1.44–1.51)	0.25	(−0.45–2.28)

MVPA: moderate to vigorous physical activity. *ß:* Standardised beta coefficient. ** Significant association in lineal regression (*p* < 0.001).

**Table 4 ijerph-17-06864-t004:** Associations between parents’ mode of commuting to work and their children mode of commuting to school by gender.

	Mother		Father			Mother		Father	
	OR	95% CI	OR	95% CI		OR	95% CI	OR	95% CI
**Children ACS**					**Adolescents ACS**				
Passive (Ref.)	1		1		Passive (Ref.)	1		1	
All Active	4.96 **	(2.68–9.17)	3.64 **	(1.92–6.91)	All Active	2.11	(0.92–4.82)	2.27	(0.87–5.92)
Girls active	4.48 **	(2.01–9.96)	3.69 *	(1.83–7.47)	Girls active	2.27	(0.79–6.53)	2.52	(0.60–10.66)
Boys active	5.19 *	(1.92–14.05)	3.69	(0.65–21.09)	Boys active	1.76	(0.45–6.87)	2.02	(0.49–8.33)
**Public commuting**					**Public commuting**				
AC + Private (Ref.)	1		1		AC + Private (Ref.)	1		1	
All public	1.76	(0.47–6.67)	6.28 *	(1.38–28.53)	All public	3.77 *	(1.79–8.34)	5.28 **	(2.14–13.00)
Girls public	1.29	(0.26–6.46)	12.62 *	(2.08–76.55)	Girls public	4.59 *	(1.76–11.94)	6.30 *	(2.21–17.94)
Boys public	4.34	(0.35–53.25)	NA	NA	Boys public	2.28	(0.48–10.90)	4.20	(0.67–26.52)
**Private commuting**					**Private commuting**				
AC + Public (Ref.)	1		1		AC + Private (Ref.)	1		1	
All private	3.19 **	(2.07–4.93)	4.09 **	(2.66–6.30)	All private	6.03 **	(2.99–12.18)	9.85 **	(4.65–20.88)
Girls private	2.24 *	(1.32–3.80)	3.08 **	(1.83–5.16)	Girls private	5.50 **	(4.46–12.43)	12.72 **	(5.01–32.31)
Boys private	6.26 **	(2.84–13.81)	7.96 **	(3.48–18.20)	Boys private	17.21 *	(2.60–113.70)	5.08 *	(1.35–19.08)

OR: odds ratio adjusted for parents’ educational level and child age. CI: confidence interval; ACS: active commuting to school. AC: active commuting. NA: not available data for this group. * Statistical association in bivariate regression equation (*p* < 0.05). ** Statistical association in bivariate regression equation (*p* < 0.001).

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
