# Peer review of "Are the Parents’ and Their Children’s Physical Activity and Mode of Commuting Associated? Analysis by Gender and Age Group"

_ijerph, 2020, doi:10.3390/ijerph17186864_

Round 1

Reviewer 1 Report

The title of article does not reflect the main purpose of the research, as I understand it.

It would probably make sense to analyze only the connections parents’ physical activity separately for mothers and fathers and children and adolescents’ physical activity.

The sample in abstract should described more carefully including age of children. 

The article contains a good overview of works on the topic. Nevertheless, it is better to generalize them first and formulate a hypothesis. The main hypothesis has not formulated.

27 Conclusions: does not match the results of an empirical study. Conclusions must divided into scientific and practical parts.

165 Better - IBM SPSS Statistics (v21)

168-172. It is not clear why this associations should be analyzed. Or these connections need to be justified and interpreted. In my opinion, they are banal.

244-245. Not exactly a true statement. The role of peers among adolescents is high.

  1. Controversial statement: «preferences of young people to use passive MC could be transferred to adult life». They may well lead an active lifestyle in adulthood.

It does not matter how parents and children get to work, to school, and back. For example, for many Russian children and teenagers, it is prestigious to bring them by car. Many of them live far from schools. But at the same time, they, like their parents, can devote a lot of time to sports and, for example, ride a bike, visit a swimming pool, outdoor sports grounds, etc.

  1. Figure 1. It is not clear why mode of commuting is the important indicator of family activity. It is better to expand this classification by taking into account the activity of fathers and mothers, as well as other household members. For example, siblings and grandparents can be very active physically, and children can imitate them.
  2. Conclusion about “low association between the parents-offspring’s MVPA must checked on other samples and in other countries. The result of many reaches that fewer girls play sports than boys do, probably, related to gender stereotypes more than to the lifestyles of their parents. Such a variable as the family structure (single parent, extended or stepfamily etc.), health, a professional status of parents etc. are important for the analysis of factors of physical activity of boys and girls.

Author Response

Response to reviewers

Manuscript ID: ijerph-922793

REVIEWER 1

Comment 1:

The title of article does not reflect the main purpose of the research, as I understand it.

Response: Thank you for the comment. We have revised it to clearly identify the main purpose and we have provided a new title, as follow:

  • Previous title: “Association between parents’ physical activity and mode of commuting with their offsprings”
  • New title: “Are the parent´s and their children´s physical activity and mode of commuting associated?: analysis by gender and age group”

To clarify the title and the manuscript, we have changed the term “offspring” for children because is a more frequent term in the scientific literature.

Comment 2:

It would probably make sense to analyze only the connections parents’ physical activity separately for mothers and fathers and children and adolescents’ physical activity.

Response: We agree with the reviewer because showing separately the associations regarding gender (mother and fathers; boys and girls) and age (children and adolescents) is a priority in the current manuscript. Consequently, in the Table 3, the results of physical activity are presented separately for mothers and fathers, boys and girls, and children and adolescents. In addition, several paragraph of the discussion section explained how this relation is different according to parental gender o children’s age

Comment 3:

The sample in abstract should described more carefully including age of children. 

Response: Thank you for the comment. New information has been added in section as follow: This cross-sectional study included 686 mothers and fathers (43.4 ± 6.5 years old) and their children (children 9.7 ± 1.7 y. and adolescents 14.0 ± 1.7 y.)

Comment 4:

The article contains a good overview of works on the topic. Nevertheless, it is better to generalize them first and formulate a hypothesis. The main hypothesis has not formulated.

Response: Thank you for the suggestion. A hypothesis has been added to the introduction section as follow: “In the current study, it is expected to verify the association between parent-child PA and to find new associations between the AC that support the importance of the family in the development of active behaviours”.

Comment 5:

27 - Conclusions: does not match the results of an empirical study. Conclusions must divide into scientific and practical parts.

Response: Thank you your very good suggestion. We agreed and the sentence has been rewritten. The limited number of words in the abstract (200 words) does not allow for further clarification. However, these have been included in the conclusions at the end of the manuscript.

Comment 6:

165 - Better - IBM SPSS Statistics (v21)

Response: Thank you. The sentence has been modified.

Comment 7:

168-172 - It is not clear why this associations should be analyzed. Or these connections need to be justified and interpreted. In my opinion, they are banal.

Response: Thank you for the comment. We agree with the reviewer that these connections are banal and the paragraph about this information has been removed.

Comment 8:

244-245 - Not exactly a true statement. The role of peers among adolescents is high.

Response: Thank for the comment that is highly right. The sentence has been rewritten as follow: “Additionally, children as they grow up can have active behavior independent of their parents and more related to their peers”.

Comment 9:

289 - Controversial statement: «preferences of young people to use passive MC could be transferred to adult life». They may well lead an active lifestyle in adulthood.

It does not matter how parents and children get to work, to school, and back. For example, for many Russian children and teenagers, it is prestigious to bring them by car. Many of them live far from school. But at the same time, they, like their parents, can devote a lot of time to sports and, for example, ride a bike, visit a swimming pool, outdoor sports grounds, etc.

Response: Thank you very much for the comment.

Indeed, the realities of the countries are very diverse. For example, in American countries (South America and North America) despite the fact that the distances are long, and they commute to school by car [1,2,3,4]. Perhaps in some developed countries it may happen that the lack of active commuting is compensated with extracurricular physical activity. However, we do not find any studies about it. Those countries where trips are longer are those that least meet the PA recommendations [5].

On the other hand, it is important to note that a child who is inactive is strange than becomes active when they are an adult. The literature defines that an active child is projected as an active adult. Likewise, with the inactive [6,7,8,9]. However, in this sentence only has been explained “to use passive modes of commuting could be transferred to adult life”. This like any other behavior that is carried over from youth to adulthood. We believe that this statement is correct.

Thank you for the comment. This discussion is very positive and enriching.

  1. Rodríguez-Rodríguez, F., Cristi-Montero, C., Celis-Morales, C., Escobar-Gómez, D., & Chillón, P. (2017). Impact of distance on mode of active commuting in Chilean children and adolescents. International journal of environmental research and public health14(11), 1334.

  1. Villa-González, E., Huertas-Delgado, F. J., Chillón, P., Ramírez-Vélez, R., & Barranco-Ruiz, Y. (2019). Associations between active commuting to school, sleep duration, and breakfast consumption in Ecuadorian young people. BMC public health19(1), 85.

  1. Reis, R. S., Hino, A. A., Parra, D. C., Hallal, P. C., & Brownson, R. C. (2013). Bicycling and walking for transportation in three Brazilian cities. American journal of preventive medicine44(2), e9-e17.

  1. Rodríguez-Rodríguez, F., Pakomio Jara, O., Kuthe, N. M., Herrador-Colmenero, M., Ramirez-Velez, R., & Chillón, P. (2019). Influence of distance, area, and cultural context in active commuting: Continental and insular children. PloS one14(3), e0213159.

  1. Peralta, M.; Henriques-Neto, D.; Bordado, J.; Loureiro, N.; Diz, S.; Marques, A. Active Commuting to School and Physical Activity Levels among 11 to 16 Year-Old Adolescents from 63 Low- and Middle-Income Countries. Int. J. Environ. Res. Public Health 2020, 17, 1276.

  1. Telama, R., Yang, X., Laakso, L., & Viikari, J. (1997). Physical activity in childhood and adolescence as predictor of physical activity in young adulthood. American journal of preventive medicine13(4), 317-323.

  1. Telama, R., Yang, X., Hirvensalo, M., & Raitakari, O. (2006). Participation in organized youth sport as a predictor of adult physical activity: a 21-year longitudinal study. Pediatric Exercise Science18(1), 76-88.

  1. Poulsen, P. H., Biering, K., & Andersen, J. H. (2015). The association between leisure time physical activity in adolescence and poor mental health in early adulthood: a prospective cohort study. BMC Public Health16(1), 3.

  1. Rovio, S. P., Yang, X., Kankaanpää, A., Aalto, V., Hirvensalo, M., Telama, R., ... & Tammelin, T. H. (2018). Longitudinal physical activity trajectories from childhood to adulthood and their determinants: the young Finns study. Scandinavian journal of medicine & science in sports28(3), 1073-1083.

Comment 10:

306 - Figure 1. It is not clear why mode of commuting is the important indicator of family activity. It is better to expand this classification by taking into account the activity of fathers and mothers, as well as other household members. For example, siblings and grandparents can be very active physically, and children can imitate them.

Response: Thank you for the comment. The classification has been explained in more detail in the text as follow:

“According to the classification, our proposal defines to “Active Family” those parents and children that meet the MVPA recommendations and use AC. “Moderately Active Family”, those who use AC, but parents or children do not meet MVPA recommendations. In this case, the categorization is affected when any member does not meet the MVPA recommendations. “Inactive Family”, the families where neither parents nor children meet the MVPA recommendations. It makes the entire group inactive, even though some member actively commutes. Finally, “Inactive and Passive Family”, that includes the families where parents and children do not meet MVPA recommendations and use passive commuting.

For a better interpretation it should be considered that other members may positively or negatively affect PA or AC (i.e., grandparents, siblings, uncles, etc.). Therefore, by pointing to "parent", it may refer to any parental figure influencing children”.

Comment 11:

319 - Conclusion about “low association between the parents-offspring’s MVPA must checked on other samples and in other countries. The result of many reaches that fewer girls play sports than boys do, probably, related to gender stereotypes more than to the lifestyles of their parents. Such a variable as the family structure (single parent, extended or stepfamily etc.), health, a professional status of parents etc. are important for the analysis of factors of physical activity of boys and girls.

Response: Thank you for the comment. We agree, because many factors can influence in the lack association in MVPA. A new paragraph has been added to explain these variables, as follow:

“The results in the current study have demonstrated a weak association between the parent-child MVPA. This lack of association could be related to gender stereotypes, family structure (single parent, extended or reconstituted family, etc.), social problems, among other factors. More studies should be conducted to further explore the relationships and associations between parent-child MVPA in different countries and realities”.

Reviewer 2 Report

Specific and general comments are below:

Line 27 – I do not think that your findings support that statement. Your study is not even about school based introductions, so this statement cannot be made.

36: use small instead of "little"

36 - clarify sentence that begins on this line

44- replace "and" with "but"

51- clarify sentence that begins on this line

use AC and ACS consistently

77 reword sentence that begins on this line

I would recommend using the work "children" instead of "offspring or offsprings"

 - were your really allowed to get the home addresses of all subjects? I can’t imagine that being allowed

143 – the word “along” does not fit there

213 – reword this sentence; tell the reader immediately why your results are interesting/important

237 – do you know that the fathers worked long hours?

253 – replace enhance with “investigate”

299 – means

299 reword sentence that begins on this line

319 – replace low with “weak”

In the discussion put more effort into explaining the inverse relationships.

Author Response

Response to reviewers

Manuscript ID: ijerph-922793

REVIEWER 2

Reviewer comments

Specific and general comments are below:

Comment 1:

Line 27 – I do not think that your findings support that statement. Your study is not even about school based introductions, so this statement cannot be made.

Response: Thank you for the comment. We agree with your suggestion and we have decided to rewrite the sentence as follow:

“A weak association in parent-child MVPA but a strong association in MC between parent-child was found”.

Comment 2:

36- use small instead of "little"

Response: Thank you, “little” has been changed for “small”.

Comment 3:

36 - clarify sentence that begins on this line

Response: Thank you, the sentence has been improved, as follow:

“Unfortunately, a small proportion of children/adolescents [5, 6] and adults [7] meet these recommendations”.

Comment 4:

44- replace "and" with "but"

Response: Thank you, “and” has been changed by “but”.

Comment 5:

51- clarify sentence that begins on this line use AC and ACS consistently

Response: Thank you for the comment. The sentence has been corrected. Same the consistent use of AC and ACS in whole text.

Comment 6:

77- reword sentence that begins on this line.

I would recommend using the work "children" instead of "offspring or offsprings" - were your really allowed to get the home addresses of all subjects? I can’t imagine that being allowed.

Response: Thank you for the comment. The sentence has been corrected. The word offsprings was chosen because explained better both (children and adolescents). However, “offsprings” has been deleted and replaced by “children” or “child” according the sentence.

In relation to the address, it has been reported by parents in the questionnaire according to ethical permissions.

Comment 7:

143 – the word “along” does not fit there

Response: The word “along” has been deleted and the sentence was rewritten.

Comment 8:

213 – reword this sentence; tell the reader immediately why your results are interesting/important.

Response: Thank you for your comment. We added the objectives to remind them but the sentence has been removed and explained directly the main findings, according your comment.

Comment 9:

237 – do you know that the fathers worked long hours?

Response: Unfortunately, this question was not included in the questionnaire, because not has been considered as a trouble yet. However, in future studies we could be add as a variable.

This point has been included in a new section called "Study application and projections" (point 4.5).

Thank you for the comment.

Comment 10:

253 – replace enhance with “investigate”

Response: Thank you for the comment. The word has been replaced.

Comment 11:

299 – means

Response: Thank you, corrected.

Comment 12:

299 reword sentence that begins on this line

Response: The paragraph has been rewritten.

Comment 13:

319 – replace low with “weak”

Response: Thank you, the word has been replaced.

Comment 14:

In the discussion put more effort into explaining the inverse relationships.

Response: Thank you for the comment. The inverse associations have been explained better and added new information (lines 235-245).